# The Influence of Genetic Polymorphisms in Cytochrome P450 (CYP1A1 and 2D6) Gene on the Susceptibility to Philadelphia Negative Chronic Myeloid Leukemia in Sudanese Patients

**DOI:** 10.3390/ijms252413493

**Published:** 2024-12-17

**Authors:** Abozer Y. Elderdery, Hadeil M. E. Idris, Saud Nahar L. Alruwaili, Entesar M. Tebien, Abdullah Alsrhani, Fawaz O. Alenazy, Badr Alzahrani, Emad Manni, Ahmed M. E. Elkhalifa, Jeremy Mills

**Affiliations:** 1Department of Clinical Laboratory Sciences, College of Applied Medical Sciences, Jouf University, Sakaka 42421, Saudi Arabia; 2Department of Clinical Laboratory Science, College of Applied Medical Science, Shaqra University, Shaqra 15572, Saudi Arabia; 3Regional Laboratory, Ministry of Health, Al-Jouf, Sakaka 72345, Saudi Arabia; 4Department of Public Health, College of Health Sciences, Saudi Electronic University, Riyadh 13316, Saudi Arabia; 5School of Medicine, Pharmacy and Biomedical Sciences, University of Portsmouth, Portsmouth PO1 2DT, UK; jeremy.mills@port.ac.uk

**Keywords:** cytochrome P450, Ph-ve CML, CYP450, CYP1A1, CYP2D6, SNPs, Sudanese population

## Abstract

The most frequent type of leukemia in Africa is chronic myeloid leukemia (CML). The genetic background of the rarer Philadelphia chromosome (Ph) Ph-ve (BCR-ABL-ve) subform of CML is largely unknown in African patients. Therefore, in this study, we aimed to investigate the role of CYP1A1 and 2D6 SNPs in the pathogenesis of Ph-ve CML in the Sudanese population. A total of 126 patients were selected for analysis. DNA was isolated from Ph-ve CML patients and a control group for PCR-RFLP analysis of SNPs CYP1A1*2C and CYP2D6*4. The CYP1A1 gene significantly expressed the GG variant genotype (*p* < 0.05) in 23.1% of the Ph-ve CML patients and 8% of the control group. In contrast, the CYP2D6 GA genotype was strongly associated with a reduced risk of developing Ph-ve CML (*p* < 0.005) with a frequency of 50% in Ph-ve patients and 93% in the control group. CYP1A1 GG polymorphism was prevalent among patients with Ph-ve CML, suggesting its role in disease development. CYP2D6 GA (IM) polymorphism was uncommon among patients, compared with the control group, possibly indicating a protective role of the polymorphisms from Ph-ve CML. This study demonstrates an association between key metabolic SNPs and Ph-ve CML and highlights the role that altered xenobiotic metabolism may play in the development of several human leukemias.

## 1. Introduction

Chronic myeloid leukemia (CML) is a myeloproliferative neoplasm, characterized by dysregulated production and uncontrolled proliferation of both maturing and mature granulocytes but showing relatively normal differentiation. Also, a well-known feature considered a diagnostic hallmark is Philadelphia chromosome positivity. It is also known as chronic myelocytic, chronic myelogenous, or chronic granulocytic leukemia [1,2]. Still, other studies highlighted the presence of an alternative disease characterized by clinical heterogeneity and poor prognostic outcomes, defined as ‘BCR-ABL1 negative CML’ [i.e., –ve Philadelphia chromosome], which represents about 5% of total CML cases [3]. CML has a global incidence rate of 0.6–2.8 instances per 100,000 individuals. This rises significantly with age, and occurrence is more frequent in males than females, with ratios of 1.2–1.8 [4]. CML was originally considered triphasic, comprising three distinct stages: chronic, accelerated, and blast crisis. However, the prognosis for CML patients significantly improved following the introduction of tyrosine kinase inhibitors (TKIs), with under 10% of those diagnosed in the chronic phase experiencing disease progression during treatment [5]. Consequently, owing to the low incidence of CML progression and the fact that its behavior could now be considered biphasic, the new WHO 2022 classification deemed the accelerated phase less pertinent and recommended its omission [5,6].

CML diagnosis requires identification of a Ph chromosomal abnormality, in particular t(9;22) (q34;q11) translocation. This is performed with cytogenetic methods such as FISH (Fluorescence In Situ Hybridization), which has a false positive rate of 1–5%, or a molecular technique such as Reverse Transcriptase-Polymerase Chain Reaction (RT-PCR) [7]. Additionally, bone marrow aspiration is essential for diagnosis verification, assessment of disease stage, and identification of anomalies in blast and basophil percentages. It also facilitates identification of clonal evolution, such as trisomy 8 or isochromosome 17, both associated with a worse prognosis [6,7]. The predictive scores for CML (namely Sokal, Euro, and EUTOS) forecast patient outcomes from initial characteristics including age, spleen size, and blood counts. These scores are especially significant in the context of tyrosine kinase inhibitors (TKIs), which have increased survival rates but result in an increasing percentage of deaths not attributable to CML [8].

The BCR-ABL1 (Philadelphia chromosome) fusion gene is formed when the ABL gene from chromosome 9 joins with the BCR gene on chromosome 22 to create the BCR-ABL fusion gene, which can produce a variety of oncoproteins [9]. They are proteins encoded by oncogenes, either mutated or overexpressed in cancer cells. These types of proteins promote or inhibit apoptosis and uncontrolled cell growth and contribute to tumorigenesis, playing key roles in cancer progression [9,10].

Despite the initial similarity between the two subforms of CML (Ph-ve and +ve) in its initial stages, the absence of this molecular defect plays a significant role in the prediction of the course of the disease and the prognosis of Ph-ve CML; however, CML patients have favorable responses to TKIs, despite of the translocation type or Philadelphia chromosome status [11]. CML patients exhibit many mutations occur in the BCR-ABL1 kinase domain, the most notable being T315I in the ATP-binding pocket, which renders CML patients resistant to most TKIs, lowering their prognosis and survival [12]. Further mutations include P-loop mutations such as G250E, M244V, L248V, E255K, and Y253H, which impair TKI binding, while A-loop mutations like M351T and F359V lead to treatment resistance. Compound and polyclonal mutations, in particular T315I combined with others, respond poorly to treatment [12,13]. Atypical Chronic Myelogenous Leukemia (aCML), characterized by BCR/ABL1 negative and Ph chromosome-negative, is a rare and aggressive leukemia variant with poor prognosis, and although it does not exhibit the conventional CML indicators, it possesses characteristics such as hyperleukocytosis and dysgranulopoiesis. Patients with aCML are at significant risk of developing Acute Myelogenous Leukemia (AML), particularly with mutations in genes such as SETBP1, NRAS, and ASXL1 [6,9,14].

Cancer predisposition and CML have been linked to genetic variations in xenobiotic enzymes’ capacity to eliminate foreign agents; they consist of phase I enzymes (Cytochrome p450), Phase II enzymes (Glutathione-s-transferases, N-acetyltransferases), and Phase III systems (multidrug resistance-associated proteins) [15]. Cytochrome P450 (CYP450), a member of a gene superfamily in humans composed of at least 115 genes (57 functional genes and 58 pseudogenes), is responsible for the metabolism of both endogenous and exogenous entities and can convert xenobiotics or procarcinogens into DNA-reactive metabolites [16]. The cloning and characterization of human CYP450 enzymes located in the epithelium of the small intestines and smooth endoplasmic reticulum of hepatocytes revealed both genetic polymorphisms and linked variant phenotypes, which have been suggested to influence individual susceptibility to cancers as genetic modifiers of cancer risk [17]. The CYP450 complex mainly acts in oxidative catalysis and is crucial in phase I metabolic pathways of 80% of currently prescribed drugs, including cytotoxic and anticancer medications. In addition, it is involved in the synthesis of various hormones and can affect hormone-related cancers [18].

The genetic variability of drug transporters and metabolizing enzymes may affect drug response. The incidence and impact of this genetic variability differs significantly among ethnic groups, thus having an influence on drug selection and appropriate dosage calculations in different populations [19]. The cytochrome P450 1A1/2 (CYP1A1/2) enzymes are mono-oxygen enzymes that perform an essential function in the metabolism of endogenous substrates and drug breakdown, and in activating several environmental pollutants and toxins [20]. They are also known for transforming aromatic polycyclic hydrocarbon compounds, such as benzo[a]pyrene produced from smoking tobacco, into DNA-binding carcinogens [21]. CYP1A1*2C SNP A4889G (rs1048943; exon 7) leads to the substitution of Isoleucine (Ile) by Valine (Val) at position 462 in the protein, which results in a two-fold increase in catalytic activity and mutagenicity because of its increased hydrophobicity. 

Cytochrome P450 2D6 (CYP2D6) is one of the most important enzyme systems involved in drug metabolism; many reports highlight polymorphisms in the CYP2D6 gene, which impact its activity [22]. Genetic polymorphisms of CYP2D6 cause a large individual variation in enzyme activity [23]. Nearly 25% of the drugs on the market are metabolized by the non-inducible enzyme CYP2D6, and genetic variations in this enzyme might result in higher plasma levels of harmful side effects or decreased treatment effectiveness [24]. Classical substrates for CYP2D6 are lipophilic medications, which include antipsychotics, some antidepressants, anti-arrhythmic, beta-blockers, anti-emetics, and opioids [25]. Its activity is within populations to form poor metabolizer (PM), intermediate metabolizer (IM), extensive metabolizer (EM), and ultra-rapid metabolizer (UM) phenotypes [26]. This study intends to explore the prevalence of genetic polymorphisms in cytochrome (1A1 and 2D6) and their association with susceptibility to Ph-ve CML in Sudanese patients.

## 2. Results

The genotype and allele frequencies of CYP1A1*2C [rs1048943: A > G] and CYP2D6*4 polymorphisms among patients and controls and their association with the Ph-ve CML are shown in Table 1. For CYP1A1, the frequency of the homozygous *2C variant genotype (GG) was significantly higher in patients (23.1%) than in controls (8%), indicating that individuals carrying this genotype are more likely to increase the risk of developing Ph-ve CML (OR = 4.625, 95% CI = 1.181–18.119, *p* value = 0.028). No association related to heterozygous genotype (AG) was noted as it was equally distributed between patients (53.8%) and controls (55%) (OR = 1.570, 95% CI = 0.553–4.455, *p* value = 0.397). Moreover, the homozygous wild-type *1 genotype (AA) was detected in 23.1% of the patients and 37% of the controls. 

The most frequent genotype of CYP2D6 found among patients was heterozygous *4/*1 GA (50%) compared with (93%) in the control group, followed by homozygous *4/*4 variant type AA (34.6%) compared with (6%) among controls. The least frequent genotype observed in both patients and controls was homozygous wild-type GG (15.4% and 1%, respectively). Interestingly, the GA genotype which represents the IM was strongly associated with reduced risk of developing Ph-ve CML (OR = 0.035, 95% CI: 0.004–0.337, *p* value = 0.004), while no significant difference was observed in AA (PM) genotype (OR = 0.375, 95% CI = 0.033–4.228, *p* value = 0.427).

The relationships between genotypes of CYP1A1 and CYP2D6 and the demographic data of the patients including gender, age groups, and ethnic origin were also considered. No statistical differences were observed for CYP1A1 genotypes with respect to the above parameters (*p* value > 0.05). However, the only significant difference in CYP2D6 genotypes was reported with respect to gender, in which the homozygous wild type genotype (GG) and heterozygous genotype (GA) were significantly increased among females (11.5% and 30.8%) compared to males (3.8% and 19.2) (*p* value < 0.006). Interestingly, the *4/*4 variant genotype (AA) was only observed among males (34.6%) (see Table 2).

Figure 1 represents the PCR-RFLP analysis of the CYP1A1 gene polymorphism, using the BsrD1 restriction enzyme. Lane 7 shows the DNA Ladder (50 bp) and lane 6 displays the homozygous wild type (A/A), with two bands at 149 and 55 bp. Lane 3 shows the homozygous mutant type (G/G), with a single band at 204 bp. Lanes 1, 2, 4, and 5 display the heterozygous (A/G) genotype, with three bands at 204, 149, and 55 bp. Figure 2 represents the PCR-RFLP analysis of the CYP2D6 gene polymorphism using the BstN1 restriction enzyme. Lane 4 shows the DNA Ladder (50 bp) and lane 9 shows the homozygous wild type (G/G), with two bands at 104 and 230 bp. Lanes 1, 2, and 3 show the homozygous mutant type (A/A), with a single band at 334 bp. Lanes 5, 6, 7, 8, and 10 show the heterozygous (G/A) genotype, with three bands at 334, 230, and 104 bp.

## 3. Discussion

The relationships between genetic susceptibility and environmental factors in the development of various cancer types have been extensively studied [27,28]. The CYP450 metabolizing enzymes represent an important line of defense against xenobiotics and endogenous compounds, including carcinogens and therapeutic drugs [29]. However, the wide range of polymorphisms found within the CYP450 family represents a source of variation in individual susceptibility to chemically induced carcinogenesis and other diseases [30]. There are several reported studies of associations between CYP1A1 and CYP2D6 polymorphisms and the risk of developing various diseases [31]. Specifically, the CYP1A1 *2C polymorphism has been linked to an increased susceptibility to chronic obstructive pulmonary disease (COPD) and chronic kidney disease [32]. However, Peng and co-workers found that individuals with CYP1A1 2C polymorphism are less likely to develop coronary artery disease (CAD) [33]. CYP2D64 polymorphism has also been associated with an elevated risk of Parkinson’s disease, Alzheimer’s disease, and systemic sclerosis [34].

The association between CYP1A1*2C and colorectal cancer has been investigated in many studies; however, the results are controversial [35]. Increasing evidence has linked CYP1A1*2A A2455G MspI1 polymorphism with renal cell carcinoma, esophageal, lung, colorectal, and breast cancers with the prevalence of Ph+ve CML [35,36,37]. In contrast, the prevalence of the mutant allele CYP1A1*2C in an Egyptian study was 3.3% in Ph+ve CML patients and 45% in controls (OR = 0.042, 95% CI: 0.005–0.373; *p* = 0.001) [38]. To our knowledge, no studies have been carried out to investigate the differences between the polymorphism of CYP450 genotypes in both Ph-ve CML and healthy individuals. The present study aims to explore the influence of genetic polymorphisms of CYP1A1*2C and CYP2D6*4 on the susceptibility of patients with Ph-ve to CML, in Sudan.

We found that the frequency of the homozygous mutant CYP1A1 genotype (GG) was significantly higher in Ph-ve patients (23.1%) compared to controls (8%), indicating that individuals carrying it are more susceptible to developing Ph-ve CML (OR = 4.625, 95% CI = 1.181–18.119, *p* = 0.028). Several studies align with our findings that CYP1A1*2C is higher in CML than in controls. For example, a study by Lakkireddy et al. [39] demonstrated that the frequency of the GG genotype was found significantly higher among patients with Ph+ ve CML. Furthermore, a 2020 Sudanese study found that the frequency of the genotype GG was more common in CML patients than in the control group [40]. However, the CYP1A1 GG polymorphism exhibited protective properties in AML patients [35]. While some studies have suggested an association between the CYP1A1 GG genotype and increased risk of leukemia, the mechanism by which this occurs is not clear. It is possible that the increased activity of the CYP1A1 enzyme may lead to the production of reactive metabolites that can damage DNA and promote the development of cancer. However, more research is needed to fully understand the relationship between CYP1A1 GG genotype and leukemogenesis.

In this study, there was no significant association between the occurrence of the CYP1A1*2C heterozygous genotype (AG) and Ph-ve patients, as it was equally distributed between the two groups (53.8%) and (55%), respectively, (OR = 1.570, 95% CI = 0.553–4.455, *p* = 0.397). Moreover, the homozygous wild-type genotype (AA) was detected in 23.1% of CML patients and 37% of the controls. Numerous research studies have indicated that CYP1A1*2C polymorphisms are linked to increased incidence of malignancies of the head and neck, prostate cancer, and esophageal cancer [14,41,42], whilst others showed no association between CYP1A1*2C polymorphisms and gastric cancer and colorectal cancer [43,44]. Additionally, it has been reported that the effects of the polymorphism vary considerably based on race/ethnicity; Asians have a higher chance of developing esophageal cancer but Caucasians do not, and Brazilians have a five-fold rise in colon cancer compared to a two-fold rise in Caucasians [44]. These conflicting results suggest that CYP1A1 polymorphisms may have a diverse impact on cancers based on their type and patient ethnic background. Finally, patients with the (AG) allele had a significantly higher risk of renal cell carcinoma among smokers, whilst non-smokers had no association between the polymorphism and developing renal cancer [31]. It is important to acknowledge that the link between CYP1A1*2C polymorphism and renal cell carcinoma was only apparent when adding smoking as a variable, suggesting that the mutated gene does not, by itself, protect from cancer, but rather may act as a preventative mechanism.

Individuals with IM or PM CYP2D6 phenotypes, which have lower metabolic activity, may not benefit properly from therapy as when patients receive a standard drug dosage, the drugs will accumulate and could eventually be transformed into carcinogens [45]. In our research, the CYP2D6 GA IM phenotype was strongly associated with a reduced risk of developing Ph-ve CML (OR = 0.035, 95% CI: 0.004–0.337, *p* = 0.004), whereas the AA leading to a PM phenotype showed no significant association with Ph-ve CML (OR = 0.375, 95% CI = 0.033–4.228, *p* value = 0.427). Similarly, a recent study demonstrated that the GA variant allele IM was linked with lower Ph+ve CML occurrence whilst there was no substantial variation in the prevalence of homozygous mutant genotype PM between cases and controls [40]. Furthermore, a 200 CML patient study, mostly Ph+ve, from Pakistan, showed that the prevalence of CYP2D6*4 IM in CML patients was significantly higher, whilst PM polymorphism showed no statistical association as only three out of the 200 patients carried the PM gene [46]. In addition, an Egyptian study found that the rates of CYP2D6 IM and PM variants were higher in CML than in controls (28% vs. 5% and 8% vs. 0%, respectively *p* = 0.004) [45]. 

Due to the impaired detoxification of substances like benzene and anticancer drugs like cyclophosphamide, PM individuals are at increased risk for leukemia and lung cancer, according to previous research demonstrating a rise in PM genotype rates in leukemia prostatic, melanoma, bladder, colorectal, and lung cancer patients [46,47,48,49]. About 20–25% of the drugs are metabolized by CYP2D6, indicating the importance of CYP2D6 in drug metabolism [49], and the inter-individual differences in drug response among patients have been attributed to polymorphisms within this cytochrome [50]. The majority of the literature supports the findings reported here that Ph+ve CML is associated with the IM variant genotype, whilst the IM allele in our study showed a protective effect in Ph-ve CML. Additionally, PM polymorphism is controversial in the literature. This might be due to the PM allele’s rare occurrence in patients and control groups alike, as seen in the aforementioned study from Pakistan [46], or because the association was not statistically significant, as seen in our study and the study conducted in Sudan [40]. This makes it challenging to draw any definitive conclusions in regard to the PM allele. 

Contrary to the majority of the aforementioned studies, which found that CYP2D6*4 IM polymorphism incidence was higher in CML (mostly Ph+ve) patients when compared with controls, our results showed a protective property of (IM) allele against Ph-ve CML as it was considerably higher in controls than Ph-ve CML. The connection between the CYP2D6 IM polymorphism and the various subtypes of CML based on the presence or lack of the Philadelphia chromosome is perplexing; additional research is required to investigate the IM pathway in Ph-ve and Ph+ve CML. Additionally, a larger sample size is needed to confirm the association between CML and PM as it was only observed infrequently in our study, nine (34.6%) times in CML and six (6%) times in controls. The importance of identifying CYP2D6 as a potential CML risk factor may aid in better predicting the outcome of treatment; as the availability of predictive biomarkers for drug response can become indispensable guidance for the proper initiation and surveillance of dosage and treatment efficacy.

In this study, patient age does not seem to influence the susceptibility to Ph-ve CML, although interestingly, all the females in the cohort are either GG (EM) or GA (IM), but none of them bear the AA associated with poor metabolizers. However, the AA genotype was only observed among males (34.6%). This finding is similar to that of Sailaja et al. [46] who found that poor metabolizers exist only among male CML patients. Moreover, the study revealed that individuals of Afro-Asiatic ethnicity were the most affected by Ph-ve CML with 22 cases (84.6%), followed by those of Nilo-Saharan ethnicity with 3 (11.5%), while those of Niger-Congo ethnicity were least prone to develop the disease with 1 (3.8%). Similar findings were observed among patients with Ph+ve CML and other types of cancer such as meningioma [51]. However, no association between genotypes of both CYP1A1 and CYP2D6 and the ethnic origin of the patients was found in this study. Still, factors like location, pattern, and duration of carcinogen exposure, carcinogen intensity, and genetic makeup must be critically analyzed when estimating the impact of CPY450 polymorphism and must be balanced against the presence of CYP450 enzyme inhibitors and gene suppressors.

## 4. Materials and Methods

This is a case-control study that was carried out at the Radiation and Isotopes Center (RICK) in Khartoum, Sudan, and consisted of 126 Sudanese individuals, who were divided into two groups: a patient group and a control group. The patient group included 26 patients (15 males and 11 females) diagnosed with Ph-ve CML based on complete blood count (CBC), bone marrow examination, and the absence of the BCR-ABL1 gene indicated by a negative RT-PCR result [6]. Patients with other chronic myeloproliferative disorders such as polycythemia vera, essential thrombocythemia, chronic neutrophilic leukemia, and precursors B cell lymphoblastic leukemia (ALL) were excluded from the study. The control group was composed of 100 healthy individuals (51 males and 49 females) with no evidence of any personal or family history of cancer. Demographic data for both groups were collected via a questionnaire. The study was conducted according to the guidelines of the Declaration of Helsinki and the protocol was approved by the Institutional Ethical Committee of Al-Neelain University (AL-18/02/10). Informed consent was obtained from all subjects involved in the study.

Peripheral blood samples were obtained from all subjects, and genomic DNA was isolated by guanidinium chloride extraction. The purified DNA samples were dissolved in Tris EDTA and kept at −20 °C until use. Genotypes for both CYP1A1*2C and CYP2D6*4 polymorphisms were detected by using the polymerase chain reaction followed by restriction digestion (PCR-RFLP), as it is still widely used in genetic mapping, species identification, and detecting mutations in specific genes (A, B) as described previously [40].

This technique combines PCR amplification of the required DNA region followed by restriction enzyme digestion and gel electrophoresis. Initially, DNA is amplified using PCR, targeting a gene or region of interest using primer pairs, as shown in Table 3. Next, restriction enzymes, BsrD1, and BstN1 were used to cut the PCR products at their specific recognition sites (also highlighted in Table 3 and Figure 3). The resulting digestion fragments were separated by gel electrophoresis producing a unique pattern of band sizes depending on the impact of the SNP on the recognition site of the enzyme. The banding patterns observed can be compared between samples and are indicative of the individual patient’s genotype [6].

All primer pairs used were purchased from Macrogen Inc. (Seoul, Republic of Korea). Following amplification, restriction digestion was performed, and fragments were observed on a 3% agarose gel using electrophoresis. Three banding patterns were observed for each gene and are listed in Table 3. 

Statistical analysis was calculated by the Statistical Package for the Social Sciences Version 25 (SPSS Inc., Chicago, IL, USA). Associations between the genotypes among patients and control were detected using a chi-square test. The disease risk was evaluated by the odd ratio (OR) with a 95% confidence interval (95% CI). *p*-values less than 0.05 were considered statistically significant. 

## 5. Conclusions

The homozygous mutation GG of CYP1A1 was significantly prevalent among Ph-ve CML patients, suggesting that it may aid in the development of the disease. The polymorphic heterozygous variant allele (IM) in CYP2D6 was uncommon among Ph-ve CML patients when compared with the control group, suggesting that the (IM) allele could potentially have a protective effect against Ph-ve CML. These alleles could potentially play an important role in the pathogenesis and treatment of Ph-ve CML. Further studies concerning Ph-ve CML in correlation with clinical symptoms and type of medications are indicated.

## Figures and Tables

**Figure 1 ijms-25-13493-f001:**
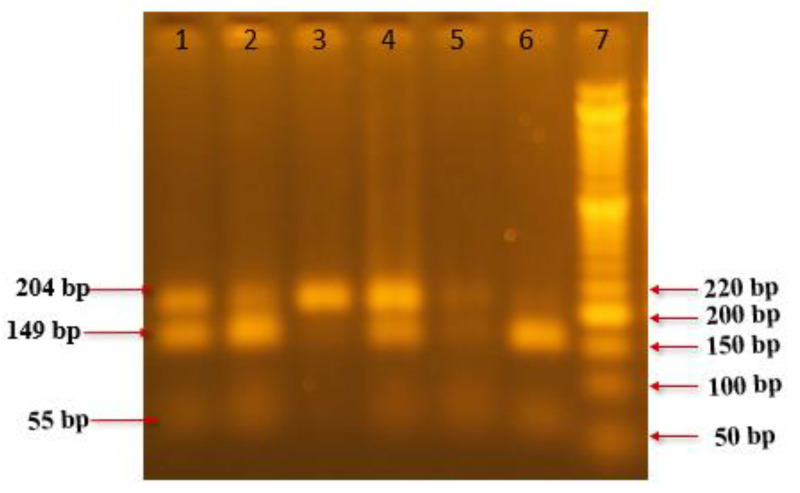
PCR-RFLP analysis of CYP1A1 gene polymorphism using BsrD1 restriction enzyme. Lane 7 shows DNA Ladder 50 bp. Lanes 6 shows Homozygous wild type (A/A) two bands at 149 and 55bp. Lanes 3 shows homozygous mutant type (G/G) one band at 204bp. Lanes 1, 2, 4, and 5 show heterozygous (A/G) three bands at 204, 149 and 55 bp.

**Figure 2 ijms-25-13493-f002:**
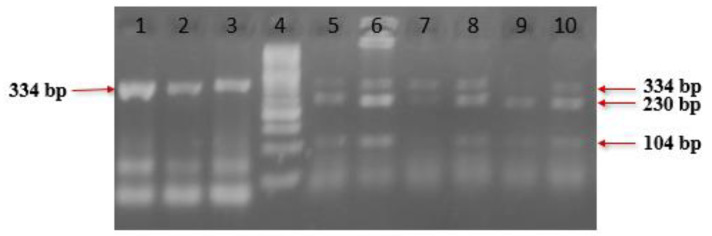
PCR-RFLP analysis of CYP2D6 gene polymorphism using BstN1 restriction enzyme. Lane 4 shows DNA Ladder 50 bp. Lanes 9 shows homozygous wild type (G/G) two bands at 104 and 230 bp. Lanes 1, 2, and 3 show homozygous mutant type (A/A) one band at 334bp. Lanes 5, 6, 7, 8, and 10 show heterozygous (G/A) three bands at 334, 230 and 104 bp.

**Figure 3 ijms-25-13493-f003:**
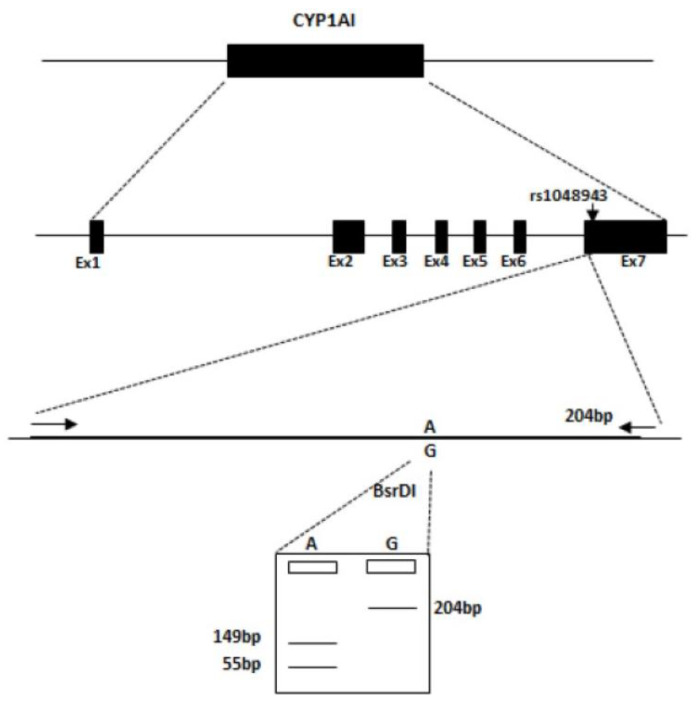
This schematic describes the steps involved in RFLP for CYP1A1*2C. The SNP CYP1A1*2C A4889G (rs1048943) presents with G in the wild type and A in the polymorphic allele. The presence of the polymorphism introduces a restriction enzyme cut-site for BsrDI, allowing identification of an individual’s genotype.

**Table 1 ijms-25-13493-t001:** Genotype and allele frequencies of CYP1A1 and CYP2D6 polymorphisms among cases and controls and their association with the Ph-ve CML.

Genotypes/Allele Frequency		Patient N (%)	Control N (%)	OR	95%CI	*p*-Value
CYP1A1*2C	AA	6 (23.1%)	37 (37%)	Reference		
	AG	14 (53.8%)	55 (55%)	1.570	0.55–4.46	0.397
	GG	6 (23.1%)	8 (8%)	4.625	1.18–18.12	0.028
	A	26 (50%)	129 (64.5%)	-	-	-
	G	26 (50%)	71 (35.5%)	-	-	-
CYP2D6*4	GG (EM)	4 (15.4%)	1 (1%)	Reference		
	GA (IM)	13 (50%)	93 (93%)	0.035	0.004–0.34	0.004
	AA (PM)	9 (34.6%)	6 (6%)	0.375	0.03–4.23	0.427
	G	21 (40.4%)	95 (47.5%)	-	-	**-**
	A	31 (59.6%)	105 (52.5%)	-	-	**-**

Key: N = total number; OR = odd ratio; CI = confidence interval; EM = extensive metabolizer (homozygous wild type status); IM = intermediate metabolizer (heterozygous status); PM = poor metabolizer (homozygous mutant status).

**Table 2 ijms-25-13493-t002:** Distribution of CYP1A1 and CYP2D6 genotypes according to demographic data of the patients.

Parameter		CYP1A1 N (%)			CYP2D6 N (%)		
		AA	AG	GG	GG(EM)	GA(IM)	AA(PM)
Gender	Male	2 (7.7%)	10 (38.5%)	3 (11.5%)	1 (3.8%)	5 (19.2%)	9 (34.6%)
	Female	4 (15.4%)	4 (15.4%)	3 (11.5%)	3 (11.5%)	8 (30.8%)	0 (0.0%)
	*p* value	0.261			0.006		
Age (Year)	<40	2 (7.7%)	1 (3.8%)	1 (3.8%)	2 (7.7%)	1 (3.8%)	1 (3.8%)
	>40	4(15.4%)	13 (50%)	5 (19.2%)	2 (7.7%)	12 (46.2%)	8 (30.8%)
	*p* value	0.329			0.111		
Ethnic groups	Afro-Asiatic	5 (19.2%)	13 (50%)	4 (15.4%)	3 (11.5%)	11 (42.3%)	8 (30.8%)
	Nilo-Saharan	1 (3.8%)	1 (3.8%)	1 (3.8%)	1 (3.8%)	1 (3.8%)	1 (3.8%)
	Niger-Congo	0 (0.0%)	0 (0.0%)	1 (3.8%)	0 (0.0%)	1 (3.8%)	0 (0.0%)
	*p* value	0.382			0.762		

Key: N = total number; EM = extensive metabolizer (homozygous wild type status); IM = intermediate metabolizer (heterozygous status); PM = poor metabolizer (homozygous mutant status).

**Table 3 ijms-25-13493-t003:** Primers, PCR product sizes, restriction enzymes, and banding patterns for CYP1A1*2C and CYP2D6*4.

Gene	PRIMERS	PCR Product	Restriction Enzyme	Banding Patterns
CYP1A1*2C	F-5′ CTG TCT CCC- TCT GGT TAC AGG AAG C-3 R-5′ TTC CAC CCG TTG CAG CAG GAT AGC C-3′	204 bp	BsrD1	*1/*1 (A/A): 149 and 55 bp *1/*2C (A/G): 204, 149 and 55 bp *2C/*2C (G/G): 204 bp
CYP2D6*4	F-5′GCT TCG CCAA CCA CTC CG-3′ R- 5′AAA TCC TGC TCT TCC GAG GC-3	334 bp	BstN1	*1/*1 (G/G EM): 230 and 104 bp. *1/*4 (G/A IM): 334, 230 and 104 bp. *4/*4 (A/A PM): 334 bp.
**Gene**	**PCR Product**	**Restriction Enzyme**		**Banding Patterns**
CYP1A1*2C	204 bp	BsrD1		*1/*1 (A/A): 149 and 55 bp *1/*2C (A/G): 204, 149 and 55 bp *2C/*2C (G/G): 204 bp
CYP2D6*4	334 bp	BstN1		*1/*1 (G/G EM): 230 and 104 bp. *1/*4 (G/A IM): 334, 230 and 104 bp. *4/*4 (A/A PM): 334 bp.

## Data Availability

The data used to support the findings of this study are included in the manuscript. All the data and information of this research are available from A Y Elderdery upon request.

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
