# Peer review of "The Influence of Genetic Polymorphisms in Cytochrome P450 (CYP1A1 and 2D6) Gene on the Susceptibility to Philadelphia Negative Chronic Myeloid Leukemia in Sudanese Patients"

_ijms, 2024, doi:10.3390/ijms252413493_

Round 1
Reviewer 1 Report
Comments and Suggestions for Authors
This study analyzed two SNPs of cytochrome P450 gene in "atypical" BCR-ABL1 negative chronic myeloid leukaemia of Sudan. I understand that not much information is available from that country. Therefore, the study is important for scientific community. Nowadays, the same SNPs may be studied using NGS, but the technique of the study may be still used. The text is well written adn easy to read.
Comments:
(1) Line 16. Please write "Chronic myeloid leukеmia" instead of "chronic".
(2) In material and methods, I am aware the methodology was described in the previous publication. However, it may be useful to the readers if the PCR-RFLP is explained and described in more detail. A diagram demonstrating the restriction sites, SNP, primers, and bandings in the agarose gel would be beneficial. Figures 1 and 2 of original images are fine, but the expected pattern and design would help.
(3) In original images Figure 2. Does the line 7 show band at 104bp?
(4) In In original images Figure 2. Do lines 1,2, and 3 have band at 104?
(5) Line 221. The cases are from RICK Sudan. However, the study is Saudi Arabia and Portsmouth UK. Is this correct?
(6) In the Introduction, lines 31 to 40. You may expand the description of CML.
For example, useful sentences are the following:
"Chronic myeloid leukеmia (CΜL; also known as chronic myelocytic, chronic myelogenous, or chronic granulocytic leukemia) is a myeloproliferative neoplasm characterized by the dysregulated production and uncontrolled proliferation of mature and maturing granulocytes with fairly normal differentiation."
"The BCR::ABL1 fusion gene results in the formation of a unique gene product, the BCR::ABL1 fusion protein."
"СМԼ accounts for approximately 15 to 20 percent of leukemias in adults"
"The prevalence of CМԼ is steadily increasing in the Western world, due to the dramatic effect of ABL1 kinase inhibitors on survival"
"CMÔ¼ has a triphasic or biphasic clinical course"
Pathological features, bone marrow biopsy, genetics, diagnosis, differential diagnosis, prognosis,
***subgroup of patients harboring the BCR::ABL1 T315I mutation, which is resistant to the majority of the currently available ΤΚΙÑ•.
*** CML risk score
*** BCR::ABL1 negative and Ph negative ---> "atypical"----> poor prognosis and transformation to AML can accur.
(7) Was acute precursor B cell lymphoblastic leukemiа (ALL) excluded?
(8) Line 38. Please describe the "oncoproteins"
(9) Based on the 2016 classification of WHO, there are several diagnostic criteria of atypical chronic myeloid leukemia, BCR-ABL1 negative. As I understand, the diagnosis was performed based on complete blood count
(CBC), bone marrow examination, and the absence of the BCR-ABL1 gene indicated by a negative Reverse Transcriptase Polymerase Chain Reaction (RT-PCR) result. Please confirm the diagnosis was based on WHO. If not, please justify it.
(10) You may comments on the ICC 2022 and WHO5 diagnostic criteria.
(11) In Tables 1 and 2. Could you please format the data as ***/*** (***%) ?
(12) In the abstract, "genetic etiology" is used. However, are these SNPs pathogenic of CML or are just a biomarker of the disease?
(13) Line 20, please correct to "PCR-RFLP" if necessary.
(14) In the abstract, line 16. Please better not using "genetic etiology", I would use "background" or similar.
(15) Line 23, "risk of developing". This study is not showing causality, but association. Please change words.
(16) Why is group AA and GG (EM) the reference in the OR and p value calculation?
(17) Abstract lacks conclusion.
(18) Did you compare with Ph+ CML?
(19) Are these SNPs associated with other diseases?
(20) Is the number of cases enough?
(21) Confirm that data and p values between text and tables is correct.
Author Response
Point-by-point reply to reviewer’s comments:
REVIEWER 1:
Comments and Suggestions for Authors
This study analyzed two SNPs of cytochrome P450 gene in "atypical" BCR-ABL1 negative chronic myeloid leukaemia of Sudan. I understand that not much information is available from that country. Therefore, the study is important for scientific community. Nowadays, the same SNPs may be studied using NGS, but the technique of the study may be still used. The text is well written and easy to read.
Reply: We thank the reviewer for this comment.
Comments:
Comment-1: Line 16. Please write "Chronic myeloid leukеmia" instead of "chronic".
Reply: Thank you for your valuable comment. It has been corrected, (Line 16).
Comment-2: In material and methods, I am aware the methodology was described in the previous publication. However, it may be useful to the readers if the PCR-RFLP is explained and described in more detail. A diagram demonstrating the restriction sites, SNP, primers, and bandings in the agarose gel would be beneficial. Figures 1 and 2 of original images are fine, but the expected pattern and design would help.
Reply: Thank you for your comment. The PCR-RFLP is explained and described in more detail, Page 7, (Line 307 - 317) with two references.
PCR-RFLP (Polymerase Chain Reaction-Restriction Fragment Length Polymorphism) is widely used in genetic mapping, species identification, and detecting mutations in specific genes (A, B). Therefore, it was used here as a molecular technique for detecting genetic variation of both CYP1A1*2C and CYP2D6*4 polymorphisms.
This technique combines PCR amplification of a certain DNA region with restriction enzyme digestion. First, DNA is amplified using PCR, targeting a gene or region of interest, as shown in Table 3. Next, restriction enzymes, BsrD1 and BstN1 were used to cut the PCR products at specific recognition sites that were mentioned in this table as well. The resulting fragments were separated by gel electrophoresis, producing a unique pattern of band sizes, which were compared between samples see new Figure 1 (page 12).
References:
- Smith G, Apperley J, Milojkovic D, Cross NC, Foroni L, Byrne J, et al. A British Society for Haematology Guideline on the diagnosis and management of chronic myeloid leukaemia. British Journal of Haematology. 2020;191(2):171-93.
- Idris HME, Khalil HB, Mills J, Elderdery AY. CYP1A1 and CYP2D6 Polymorphisms and Susceptibility to Chronic Myelocytic Leukaemia. Curr Cancer Drug Targets. 2020;20(9):675-80.
A diagram demonstrating the restriction sites, SNP, primers, and bandings in the agarose gel would be beneficial has been done in the below Figure 1, page 12.
Figure 1 – This schematic diagram describes the steps involved in RFLP for CYP1A1*2C. the SNP CYP1A1*2C A4889G (rs1048943) presents with G in the wild type and A in the polymorphic allele. The presence of the polymorphism introduces a restriction enzyme cut-site for BsrDI allowing genotypic identification of individuals.
Comment-3: In original images Figure 2. Does the line 7 show band at 104bp?
Reply: Thank you for your comment. Yes, it shows a band at 104 bp, but it is faint.
Comment-4: In original images Figure 2. Do lines 1,2, and 3 have band at 104?.
Reply: Thank you for your valuable comment. No, these lines (1, 2 and 3) have no band at 104. This band always appears when we see 230 and/ or 334 (i.e. this band appears in the digested case that indicates the presence of restriction sites in the of CYP2D6 gene (+); from which Homozygous wild type (G/G) two bands at 104 & 230 bp and Heterozygous (G/A) three bands three bands at 334, 230 & 104 bp are analysed.
Comment-5: The cases are from RICK Sudan. However, the study is Saudi Arabia and Portsmouth UK. Is this correct?
Reply: Thank you for the comment. Thank you for the comment. We would like to let you know that all co-authors did contribute to this manuscript, as stated in the author contributions section. We have run several research projects together in various locales, as I did my PhD and Postdoc at the University of Portsmouth, England (2007 - 2012), and since have been working at Jouf University from November 2013, and have also at various Sudanese Universities, teaching supervising post graduate students since 2001. We have also two co-authors, Hadeil M. E. Idris and Entisar M Tebien and I from Sudan who have been working at Sudanese Universities and moved to work for Saudi Arabia Universities.
Comment-6: In the Introduction, lines 31 to 40. You may expand the description of CML.
For example, useful sentences are the following:
"Chronic myeloid leukеmia (CΜL; also known as chronic myelocytic, chronic myelogenous, or chronic granulocytic leukemia) is a myeloproliferative neoplasm characterized by the dysregulated production and uncontrolled proliferation of mature and maturing granulocytes with fairly normal differentiation."
"The BCR::ABL1 fusion gene results in the formation of a unique gene product, the BCR::ABL1 fusion protein."
"СМԼ accounts for approximately 15 to 20 percent of leukemias in adults"
"The prevalence of CМԼ is steadily increasing in the Western world, due to the dramatic effect of ABL1 kinase inhibitors on survival"
"CMÔ¼ has a triphasic or biphasic clinical course"
Pathological features, bone marrow biopsy, genetics, diagnosis, differential diagnosis, prognosis,
***subgroup of patients harboring the BCR::ABL1 T315I mutation, which is resistant to the majority of the currently available ΤΚΙÑ•.
*** CML risk score
*** BCR::ABL1 negative and Ph negative ---> "atypical"----> poor prognosis and transformation to AML can accur.
Reply: Thank you for your comment to expand the description of CML. We have added some paragraphs with 14 refences in the introduction section as you requested:
Chronic Myeloid Leukaemia (ML) is a myeloproliferative neoplasm, characterised by dysregulated production and uncontrolled proliferation of both maturing and mature granulocytes, but showing relatively normal differentiation. Also, a well-known feature, considered a diagnostic hallmark, is Philadelphia chromosome positivity. It is also known as chronic myelocytic, chronic myelogenous, or chronic granulocytic leukemia [1, 2]. (See Line 44 -52, Page 2).
CML has a global incidence rate of 0.6 - 2.8 instances per 100,000 individuals. This rises significantly with age, and occurrance is more frequent in males than females, with ratios of 1.2 - 1.8 [4]. CML was originally considered triphasic, comprising three distinct stages: chronic, accelerated, and blast crisis. However, the prognosis for CML patients significantly improved following the introduction of Tyrosine Kinase Inhibitors (TKIs), with under 10% of those diagnosed in chronic phase experiencing disease progression during treatment [5]. Consequently, owing to the low incidence of CML progression and the effect that its behavior could now be considered biphasic, the new WHO 2022 classification deemed the accelerated phase less pertinent, and recommended its omission [5, 6]. (See Line 34 -40, Page 2).
CML diagnosis requires identification of Ph chromosomal abnormality, in particular t(9;22) (q34;q11) translocation. This is done with cytogenetic methods such as FISH (Fluo-rescence In Situ Hybridization), which has a false positive rate of 1 - 5%, or a molecular technique such as Reverse Transcriptase-Polymerase Chain Reaction (RT-PCR) [7]. Addi-tionally, bone marrow aspiration is essential for diagnosis verification, assessment of disease stage, and identification of anomalies in blast and basophil percentages. It also fa-cilitates identification of clonal evolution, such as trisomy 8 or isochromosome 17, both associated with a worse prognosis [6,7]. The predictive scores for CML (namely Sokal, Euro, and EUTOS) forecast patient outcomes from initial characteristics including age, spleen size, and blood counts; These scores are especially significant in the context of ty-rosine kinase inhibitors (TKIs), which have increased survival rates but result in an in-creasing percentage of deaths not attributable to CML [8]. (See Line 53 -64, Page 2).
They are proteins encoded by oncogenes, either mutated or overexpressed in cancer cells. These types of proteins promote inhibit apoptosis, uncontrolled cell growth and contribute to tumorigenesis, playing key roles in cancer progression [9,10]. (See Line 67 -70, Page 2).
CML patients exhibit many mutations occur in the BCR-ABL1 kinase domain, the most notable being T315I in the ATP-binding pocket, which renders CML patients resistant to most TKIs, lowering their prognosis and survival [12]. Further mutations include P-loop such as G250E, M244V, L248V, E255K, and Y253H which impair TKI binding, while A-loop mutations like M351T and F359V lead to treatment resistance. Compound and polyclonal mutations, in particular T315I combined with others, respond poorly to treatment [12,13].. Atypical Chronic Myelogenous Leukemia (aCML), characterized by BCR/ABL1 negative and Ph chromosome-negative, is a rare and aggressive leukemia variant with poor prognosis, and although it does not exhibit the conventional CML indicators, it possesses characteristics such as hyperleukocytosis and dysgranulopoiesis. Patients with a CML are at significant risk of developing Acute Myelogenous Leukemia (AML), particularly with mutations in genes such as SETBP1, NRAS, and ASXL1 [6,9,14]. (See Line 73 -86, Page 2).
Comment-7: Was acute precursor B cell lymphoblastic leukemiа (ALL) excluded?
Reply: We thank the reviewer for the valuable comments on the exclusion acute precursor B cell lymphoblastic leukaemia (ALL). We have studied this previously and therefore it has been included in the exclusion criteria, Materials and Methods section, Line 296.
Comment-8: Please describe the "oncoproteins"
Reply: Thank you for the insightful comment. It has been described, using two references.
They are proteins encoded by oncogenes, either mutated or overexpressed in cancer cells. These types of proteins promote inhibit apoptosis, uncontrolled cell growth and contribute to tumorigenesis, playing key roles in cancer progression [9,10]. (See Line 67 -70, Page 2).
Comment-9: Based on the 2016 classification of WHO, there are several diagnostic criteria of atypical chronic myeloid leukemia, BCR-ABL1 negative. As I understand, the diagnosis was performed based on complete blood count (CBC), bone marrow examination, and the absence of the BCR-ABL1 gene indicated by a negative Reverse Transcriptase Polymerase Chain Reaction (RT-PCR) result. Please confirm the diagnosis was based on WHO. If not, please justify it.
Reply: Thank you for the insightful comment. It has been included in the previous paragraphs with some references.
Comment-10: You may comments on the ICC 2022 and WHO5 diagnostic criteria.
Reply: We thank the reviewer for the valuable comments. It has been included in the previous paragraphs which we have added.
Comment-11: In Tables 1 and 2. Could you please format the data as ***/*** (***%) ?
Reply: We thank the reviewer for the valuable comments. It has been formatted.
Comment-12: In the abstract, "genetic etiology" is used. However, are these SNPs pathogenic of CML or are just a biomarker of the disease?
Reply: We thank the reviewer for the valuable comments a change has been made see comment 14.
Comment-13: Line 20, please correct to "PCR-RFLP" if necessary.
Reply: Thank you for your comment. It has been corrected
Comment-14: In the abstract, line 16. Please better not using "genetic etiology", I would use "background" or similar.
Reply: Thank you for your comment. It has been changed, using background instead of the etiology.
Comment-15: Line 23, "risk of developing". This study is not showing causality, but association. Please change words.
Reply: Thank you for your comment. It has been corrected.
Comment-16: Why is group AA and GG (EM) the reference in the OR and p value calculation?
Reply: Thank you for your comment. AA and GG are considered as wild types for CYP1A1 and CYP2D6 respectively.
Comment-17: Abstract lacks conclusion.
Reply: Thanks for highlighting this, it has now been added. Lines 28 – 30, Page 1.
This study demonstrates an association between key metabolic SNPs and Ph -ve CML and high-lights the role that altered xenobiotic metabolism may play in the development of several of human leukaemias.
Comment-18: Did you compare with Ph+ CML?
Reply: We thank the reviewer for the valuable comments. Yes, we have compared them in the discussion chapter, Lines 188 – 231.
Comment-19: Are these SNPs associated with other diseases?
Reply: We thank the reviewer for the valuable comments…we have added a paragraph in the discussion section with four refences:
There several reported studies associations between CYP1A1 and CYP2D6 polymor-phisms and the risk of developing various diseases [31]. Specifically, the CYP1A1 *2C polymorphism has been linked to an increased susceptibility to chronic obstructive pul-monary disease (COPD) and chronic kidney disease [32]. However, Peng and co-workers, found that individuals with the CYP1A1 2C polymorphism are less likely to develop cor-onary artery disease (CAD) [33]. The CYP2D64 polymorphism has also been associated with an elevated risk of Parkinson’s disease, Alzheimer's disease, and systemic sclerosis [34], Line 180 -187, Page 12.
Refences:
- Lu, Y.; Qin, X.; Li, S.; Zhang, X.; He, Y.; Peng, Q.; Deng, Y.; Wang, J.; Xie, L.; Li, T.; Zeng, Z. Quantitative assessment of CYP2D6 polymorphisms and risk of Alzheimer's disease: a meta-analysis. J Neurol Sci 2014, 343, 15-22, doi:10.1016/j.jns.2014.05.033.
- Zhu, X.; Wang, Z.; He, J.; Wang, W.; Xue, W.; Wang, Y.; Zheng, L.; Zhu, M.L. Associations between CYP1A1 rs1048943 A > G and rs4646903 T > C genetic variations and colorectal cancer risk: Proof from 26 case-control studies. Oncotarget 2016, 7, 51365-51374, doi:10.18632/oncotarget.10331.
- Zheng, H.; Zhao, Y. Association of CYP1A1 MspI polymorphism in the esophageal cancer risk: a meta-analysis in the Chinese population. Eur J Med Res 2015, 20, 46, doi:10.1186/s40001-015-0135-3.
Comment-20: Is the number of cases enough?
Reply: We thank the reviewer for the valuable comments. Yes, it is okay as the cases of Philadelphia Negative Chronic Myeloid Leukaemia is rare.
(21) Confirm that data and p values between text and tables is correct.
Reply: We thank the reviewer for the valuable comments. They have been confirmed.

Reviewer 2 Report
Comments and Suggestions for Authors
Chronic Myeloid Leukemia (CML) is a clonal myeloid disorder that in the vast majority of cases is characterized by a single genetic alteration, namely the reciprocal translocation between chromosome 9 and 22 (i.e., Philadelphia -Ph- chromosome) leading to an oncogene encoding for a chimeric tyrosine kinase, Bcr-Abl1, with constitutive activity. Less frequent are cases of Ph- CML. Over the last few decades, CML Ph+ therapy has revolutionized because of the tyrosine kinase inhibitors (i.e., Imatinib, Nilotinib, Dasatinib) selectively targeting the oncogenic kinase Bcr-Abl1. On the contrary, Ph- CML due to the lack of a druggable target, still shows a less favorable prognosis when compared to Ph+.
The manuscript titled "The Influence of Genetic Polymorphisms in Cytochrome P450 (CYP1A1 & 2D6) Gene on the Susceptibility to Philadelphia Negative Chronic Myeloid Leukaemia in Sudanese Patients" by Elderdery A.Y and colleagues explore the prevalence of genetic polymorphisms in two genes encoding for detoxifying enzymes CYP1A1 and CYP2D6 on the susceptibility to Ph- CML in Sudanese patients.
Overall, the study offers some interesting hints, nonetheless before publication some issues require to be clarified.
- Occasionally, Ph- CML cells harbor chromosomal translocations. The authors are asked to provide cytogenetic data of the subject enrolled in the study.
- In the section "Material and Methods" the authors are asked to provide the sequence of the primers used to amplify the CYP1A1 and CYP2D6 fragments, alongside the restriction enzymes used for RFLP.
- At the end of the manuscript. After the section "References" appear two figures (i.e., Figures 1 and 2) but in the main text they are not mentioned. Please clarify the issue.
- The language would benefit from a quick revision. For example in line 82 I would propose to replace "…to have CML…" with something like "…increase the risk to develop CML…". Additionally, acronyms should be spelled out when used the first time.
- References are formatted according to the journal guidelines.
- A few typos are scattered throughout the main text (e.g., the family name of authors should always start with a capital letter, improper use of square brackets instead of parentheses, etc…).
Author Response
Point-by-point reply to reviewer’s comments:
REVIEWER 2:
Comments and Suggestions for Authors
Chronic Myeloid Leukemia (CML) is a clonal myeloid disorder that in the vast majority of cases is characterized by a single genetic alteration, namely the reciprocal translocation between chromosome 9 and 22 (i.e., Philadelphia -Ph- chromosome) leading to an oncogene encoding for a chimeric tyrosine kinase, Bcr-Abl1, with constitutive activity. Less frequent are cases of Ph- CML. Over the last few decades, CML Ph+ therapy has revolutionized because of the tyrosine kinase inhibitors (i.e., Imatinib, Nilotinib, Dasatinib) selectively targeting the oncogenic kinase Bcr-Abl1. On the contrary, Ph- CML due to the lack of a druggable target, still shows a less favorable prognosis when compared to Ph+.
The manuscript titled "The Influence of Genetic Polymorphisms in Cytochrome P450 (CYP1A1 & 2D6) Gene on the Susceptibility to Philadelphia Negative Chronic Myeloid Leukaemia in Sudanese Patients" by Elderdery A.Y and colleagues explore the prevalence of genetic polymorphisms in two genes encoding for detoxifying enzymes CYP1A1 and CYP2D6 on the susceptibility to Ph- CML in Sudanese patients.
Overall, the study offers some interesting hints, nonetheless before publication some issues require to be clarified.
Comment-1: Occasionally, Ph- CML cells harbor chromosomal translocations. The authors are asked to provide cytogenetic data of the subject enrolled in the study.
Reply: We thank the reviewer for the valuable comments. The Patient group included diagnosed with Ph-ve CML based on complete blood count (CBC), bone marrow examination, and the ab-sence of the BCR-ABL1 gene indicated by a negative Reverse Transcriptase Polymerase Chain Reaction (RT-PCR) result. Patients with other chronic myeloproliferative disorders such as polycythemia vera, essential thrombocythemia, and chronic neutrophilic leukemia and precursors B cell lymphoblastic leukaemia (ALL) were excluded from the study. This has been stated in the Materials and Methods (268 – 271). In this study, we aimed to investigate the role of CYP1A1 and 2D6 SNPs in the pathogenesis of Ph-ve CML SNPs in the Sudanese population.
Comment-2: In the section "Material and Methods" the authors are asked to provide the sequence of the primers used to amplify the CYP1A1 and CYP2D6 fragments, alongside the restriction enzymes used for RFLP.
Reply: We thank the reviewer for the valuable comments. The primers used to amplify the CYP1A1 and CYP2D6 fragments have been included in Table 3 (See Materials and Methods, please).
Table 3. Primers, PCR product sizes, restriction enzymes and banding patterns for CYP1A1*2C and CYP2D6*4.
Gene |
PRIMERS |
PCR Product |
Restriction Enzyme |
Banding patterns |
CYP1A1*2C |
F-5’ CTG TCT CCC- TCT GGT TAC AGG AAG C-3 R-5’ TTC CAC CCG TTG CAG CAG GAT AGC C-3’ |
204 bp |
BsrD1 |
*1/*1 (A/A): 149 and 55 bp *1/*2C (A/G): 204, 149 and 55 bp *2C/*2C (G/G) :204 bp |
CYP2D6*4 |
F-5′GCT TCG CCAA CCA CTC CG-3′ R- 5′AAA TCC TGC TCT TCC GAG GC-3 |
334 bp |
BstN1 |
*1/*1 (G/G EM): 230 and 104bp. *1/*4 (G/A IM): 334, 230 and 104bp. *4/*4 (A/A PM): 334bp. |
Comment-3: At the end of the manuscript. After the section "References" appear two figures (i.e., Figures 1 and 2) but in the main text they are not mentioned. Please clarify the issue.
Reply: We thank the reviewer for the valuable comments. They have been mentioned in the results section (Line 154 -163).
Figure 2 represents the PCR-RFLP analysis of the CYP1A1 gene polymorphism, using the BsrD1 restriction enzyme. Lane 7 shows the DNA Ladder (50 bp) and lane 6 displays the homozygous wild type (A/A), with two bands at 149 and 55 bp. Lane 3 shows the homozygous mutant type (G/G), with a single band at 204 bp. Lanes 1, 2, 4, and 5 display the heterozygous (A/G) genotype, with three bands at 204, 149, and 55 bp. Figure 3 represents the PCR-RFLP analysis of the CYP2D6 gene polymorphism using the BstN1 restriction enzyme. Lane 4 shows the DNA Ladder (50 bp) and lane 9 shows the homozygous wild type (G/G), with two bands at 104 and 230 bp. Lanes 1, 2, and 3 show the homozygous mutant type (A/A), with a single band at 334 bp. Lanes 5, 6, 7, 8, and 10 show the heterozygous (G/A) genotype, with three bands at 334, 230, and 104 bp.
Comment-4: The language would benefit from a quick revision. For example in line 82 I would propose to replace "…to have CML…" with something like "…increase the risk to develop CML…". Additionally, acronyms should be spelled out when used the first time.
Reply: Thank you for the insightful comment. It has been done as you requested (See line 32, please).
Comment-5: References are formatted according to the journal guidelines.
Reply: Thank you for the insightful comment. All references are formatted according to the journal guidelines, using EndNote.
Comment-6: A few typos are scattered throughout the main text (e.g., the family name of authors should always start with a capital letter, improper use of square brackets instead of parentheses, etc…).
Reply: We thank the reviewer for the valuable comments. All of them have been corrected.
